# Pembrolizumab-Associated Hemophagocytic Lymphohistiocytosis in Clear Cell Renal Carcinoma: Case Report and Literature Review

**DOI:** 10.3390/reports8040256

**Published:** 2025-12-03

**Authors:** Romina Pinto Valdivia, Luis Posado-Domínguez, Maria Escribano Iglesias, Patricia Antúnez Plaza, Emilio Fonseca-Sánchez

**Affiliations:** 1Medical Oncology Department, University Hospital of Salamanca, Paseo de San Vicente, 182, 37007 Salamanca, Spain; rpintov@saludcastillayleon.es (R.P.V.); efonseca@saludcastillayleon.es (E.F.-S.); 2Institute of Biomedical Research of Salamanca (IBSAL), 37007 Salamanca, Spain; 3Pathology Department, University Hospital of Salamanca, 37007 Salamanca, Spain; 4Radiodiagnosis Service, University Hospital of Salamanca, 37007 Salamanca, Spain; pantunez@saludcastillayleon.es; 5Faculty of Medicine, University of Salamanca, 37007 Salamanca, Spain

**Keywords:** pembrolizumab, immune checkpoint inhibitors, hemophagocytic lymphohistiocytosis, immune-related adverse events, immunotherapy complications, case report

## Abstract

**Background and Clinical Significance:** Immune checkpoint inhibitors (ICIs) have transformed the management of advanced solid tumors but can trigger severe immune-related adverse events (irAEs). Among the rarest and most life-threatening is hemophagocytic lymphohistiocytosis (HLH), a hyperinflammatory syndrome driven by uncontrolled immune activation. **Case Presentation:** We report the case of an 80-year-old man with clear cell renal carcinoma with sarcomatoid features who developed secondary hemophagocytic lymphohistiocytosis (HLH) after receiving four cycles of adjuvant pembrolizumab therapy. Following four cycles of immunotherapy, he presented with persistent fever, pancytopenia, hyperferritinemia (>49,000 ng/mL), hypofibrinogenemia, and elevated soluble IL-2 receptor (>7500 U/mL), fulfilling at least five HLH-2004 diagnostic criteria. Despite treatment with high-dose corticosteroids and intravenous anakinra (100 mg every 6 h), his condition rapidly deteriorated, leading to multiorgan failure and death. **Discussion:** ICI-induced HLH is an exceptional but increasingly recognized irAE, with fewer than 30 pembrolizumab-related cases reported to date. Diagnosis is challenging due to its nonspecific presentation, which can mimic infection, hepatic toxicity, or disease progression. The pathogenesis is believed to involve excessive activation of cytotoxic T cells and cytokine storm. While established pediatric protocols (HLH-94, HLH-2004) guide management, adult cases often require individualized approaches using corticosteroids and cytokine-targeted therapies such as IL-1 or IL-6 blockade. **Conclusions:** HLH secondary to ICIs should be considered in the differential diagnosis of patients receiving immunotherapy who develop unexplained fever and cytopenia. Early recognition and prompt initiation of immunosuppressive therapy are critical to improving outcomes in this potentially fatal complication.

## 1. Introduction and Clinical Significance

Over the past decade, immunotherapy with immune checkpoint inhibitors (ICIs) has transformed the treatment of multiple solid tumors, substantially changing the prognosis of diseases traditionally associated with limited survival. These agents, which target regulatory molecules such as PD-1, PD-L1, or CTLA-4, enhance the antitumor immune response by reactivating T lymphocytes that have been previously suppressed by the tumor microenvironment [1].

Among them, pembrolizumab, an anti–PD-1 monoclonal antibody, has established itself as a key therapeutic tool in non–small cell lung cancer, melanoma, head and neck squamous cell carcinoma, and renal cell carcinoma, both in advanced disease and in high-risk early stages. Its efficacy and safety profile have led to approval in multiple indications and have replaced traditional first-line therapies in various oncologic settings.

In clear cell renal cell carcinoma, pembrolizumab has demonstrated clinical benefit in combination with tyrosine kinase inhibitors in metastatic disease [2]. In the adjuvant setting, eligibility for pembrolizumab is based on the KEYNOTE-564 criteria, which include patients with high-risk pathological features after nephrectomy (pT2 grade 4 or sarcomatoid, pT3–T4, or N+ disease) [3]. Our patient met these high-risk criteria (pT3a, sarcomatoid differentiation), thus fulfilling the indications for adjuvant pembrolizumab.

However, like other ICIs, pembrolizumab can be associated with immune-related adverse events (irAEs) which, although mostly mild or moderate, can on rare occasions be life-threatening [4]. Among these, hemophagocytic syndrome secondary to immune checkpoint inhibitors is an exceptional complication, with fewer than 100 cases reported in the literature, characterized by systemic hyperinflammation, multiorgan failure, and high mortality [5].

We present the case of a patient with clear cell renal cell carcinoma with sarcomatoid differentiation, treated with adjuvant pembrolizumab, who developed hemophagocytic syndrome with a fatal outcome despite the initiation of corticosteroid therapy and anakinra.

## 2. Case Presentation

An 80-year-old man, former smoker (pack-year index 16), with a history of atrial flutter treated with ablation and stage III chronic kidney disease. In August 2023, he was diagnosed with grade 4 clear cell renal cell carcinoma, with sarcomatoid differentiation, and a pathological stage of pT3aN0M0, following a complete resection with negative surgical margins (R0) (Figure 1 and Figure 2). After surgery, adjuvant treatment with pembrolizumab was initiated.

In late December 2023, after completing the fourth cycle of immunotherapy, the patient developed progressive asthenia and persistent low-grade fever (maximum 37.7 °C) at home. Laboratory tests revealed new-onset neutropenia, mild anemia (grade 1) initially attributed to chronic disease, and cholestatic liver enzyme abnormalities (GGT > 400 U/L, ALP > 200 U/L), along with elevated C-reactive protein (CRP 4.14 mg/dL). He was empirically treated with oral antibiotics and prednisone, without significant clinical improvement.

A few days later, he developed persistent high-grade fever, loss of appetite, progressive deterioration in his general condition, and inability to perform basic activities of daily living, leading to hospitalization. During admission, laboratory tests revealed progressive cytopenia’s: grade 4 neutropenia and grade 3 thrombocytopenia, accompanied by elevated liver enzymes and acute phase reactants (Table 1 and Figure 3). Empirical treatment with piperacillin–tazobactam and vancomycin was initiated, followed by the addition of antifungal therapy due to the absence of clinical improvement.

Despite extensive microbiological investigations, including multiple blood and urine cultures, as well as tests for cytomegalovirus, Epstein–Barr virus, tuberculosis, and fungal infections, no infectious source was identified.

Given the lack of response to broad-spectrum antibiotics, negative bacterial and viral cultures, and the absence of identifiable infectious foci on clinical examination, an immune-related toxicity was suspected. Corticosteroid therapy at a dose of 1 mg/kg was initiated, with an initially partial but favorable response. However, evening fevers persisted with spiking patterns, and cytopenias continued to worsen. During hospitalization, a thoracoabdominal CT scan revealed multiple pulmonary lesions compatible with metastatic progression, as well as splenomegaly (18 cm) (Figure 4).

Due to the persistence of fever, pancytopenia, and the absence of an identifiable infectious etiology, an extended inflammatory workup was performed, including non-routine markers such as ferritin and soluble IL-2 receptor. Notably, ferritin levels were progressively elevated, reaching 49,274 ng/mL, and soluble IL-2 receptor levels exceeded 7500 U/mL. The patient met at least five of the eight diagnostic criteria for hemophagocytic lymphohistiocytosis (HLH) according to the HLH-2004 guidelines (Figure 3).

Although bone marrow biopsy could not be performed due to the patient’s poor clinical condition, diagnostic probability was further assessed using the HScore, a validated tool for adult hemophagocytic lymphohistiocytosis developed by Fardet et al. (2014) [6]. The calculated HScore was 233 points, corresponding to an estimated 98–99% probability of HLH.

Treatment with anakinra, an interleukin-1 receptor antagonist, was initiated without improvement. The patient received intravenous anakinra at 100 mg every 6 h in combination with methylprednisolone 2 mg/kg/day. He rapidly progressed to multiorgan failure, with severe hepatic dysfunction, respiratory failure, and refractory shock, dying a few days after the clinical diagnosis of pembrolizumab-induced secondary hemophagocytic syndrome.

## 3. Discussion

Hemophagocytic syndrome (HPS), also known HLH or macrophage activation syndrome, is a rare and potentially fatal entity. It is a severe disorder of the mononuclear phagocyte system, characterized by uncontrolled proliferation and activation of macrophages and T lymphocytes, leading to a state of systemic hyperinflammation, histiocyte activation, and cytokine storm [7].

HLS can be primary (genetic in origin) or secondary to infections, drugs, neoplasms, or autoimmune diseases. Within the latter group, immunotherapy with ICIs, such as pembrolizumab, has revolutionized the treatment of solid tumors such as lung cancer, melanoma, or renal carcinoma over the past decade [8].

The diagnosis of HLH remains challenging, particularly in oncology patients, where its manifestations—fever, cytopenias, and organ dysfunction—may mimic sepsis, disease progression, or drug-induced toxicity. The lack of specific diagnostic criteria and standardized management algorithms for immune-related HLH further complicates its early recognition and treatment. According to Daver et al., fewer than 50% of adults with malignancy-associated HLH receive HLH-specific therapy, largely because of delayed diagnosis or failure to recognize the syndrome [9]. Consequently, a significant proportion of cases may go untreated or even undiagnosed, reflecting the persistent lack of awareness of this condition in adult oncology.

HLH secondary to ICIs represents an exceptionally rare but life-threatening immune-related adverse event, associated with high morbidity and mortality. In cancer patients, HLH may arise from the malignancy itself, opportunistic infections, or immune activation induced by ICIs. Although uncommon, pembrolizumab-related HLH has been reported in fewer than 100 cases to date, emphasizing its rarity and severity. To better illustrate the clinical spectrum of pembrolizumab-associated HLH, we performed a literature review through PubMed and summarized 14 cases with complete clinical data in Table 2.

Among the published cases of pembrolizumab-associated HLH summarized in Table 2, only one fatal outcome was identified—the case reported by Rossignon et al. (2024) in a patient with metastatic urothelial carcinoma, who died 28 days after hospitalization despite treatment with dexamethasone and etoposide [20]. All remaining patients experienced clinical improvement, most of them responding to high-dose corticosteroids, occasionally in combination with other immunosuppressive agents such as azathioprine [12], tocilizumab [5] or etoposide [12]. In the absence of standardized management guidelines, therapeutic decisions are typically based on physician judgment, often extrapolating from pediatric protocols (HLH-94 and HLH-2004) and combining corticosteroids, immunosuppressants, or cytotoxic agents as clinically indicated. Different therapeutic combinations can be observed in Table 2. Notably, in a recent pharmacovigilance analysis of 190 cases of ICI-related HLH, the overall mortality rate reached approximately 15% [12].

These findings support that, when recognized early, HLH related to immune checkpoint inhibitors is frequently reversible under adequate immunosuppressive therapy. Moreover, bone marrow biopsy is not indispensable for diagnosis, as many patients are clinically unstable or not suitable for invasive procedures; in such cases, diagnosis must rely on strong clinical suspicion supported by laboratory and scoring criteria, such as the HLH-2004 framework and the HScore.

Notably, the onset of HLH can occur at widely variable time points—sometimes after the very first cycle of immunotherapy, and in other cases several weeks after treatment discontinuation—underscoring the need for long-term vigilance in patients exposed to ICIs [7,11]. Across published cases of pembrolizumab-associated HLH, the median time to onset was 12 weeks (interquartile range, 1.5–27.5; range, 1–74 weeks), highlighting the broad temporal spectrum of this adverse event [12]. This unpredictable timing likely reflects the complex immune dysregulation underlying the syndrome. ICI-induced HLH appears to result from excessive immune activation, particularly involving cytotoxic CD8^+^ T cells, leading to uncontrolled cytokine release and tissue damage. Pembrolizumab, a monoclonal antibody targeting programmed cell death-1 (PD-1), plays a pivotal role in modern oncology, including both adjuvant and advanced settings for renal cell carcinoma. Under physiological conditions, PD-1 signaling limits T-cell activity to preserve self-tolerance and prevent autoimmunity; however, tumor cells may exploit this pathway through PD-L1 expression, suppressing antitumor immune responses.

By blocking PD-1/PD-L1 interaction, pembrolizumab reactivates cytotoxic T-cell function and enhances immune-mediated tumor clearance. While this mechanism underlies its therapeutic efficacy, the resulting immune overactivation may also target normal tissues, predisposing to a spectrum of immune-related adverse events. Among these, HLH remains exceptionally rare but carries a high risk of fulminant progression and death if not promptly recognized and treated.

The diagnosis of HLH in adults is particularly challenging, as its manifestations may mimic severe infections, tumor progression, or hepatic toxicities. In the setting of immune checkpoint inhibitor (ICI) therapy, diagnostic uncertainty is even greater because immune-related adverse events can present with overlapping features. Although exceedingly rare, ICI-associated HLH has been estimated to occur in fewer than 0.1% of treated patients, yet carries a high risk of rapid clinical deterioration and death if not promptly recognized (8).

Although the patient was 80 years old and had relevant comorbidities, he maintained an excellent baseline functional status, with ECOG performance status 0 and a Karnofsky score of 100, being fully independent for activities of daily living (IADL and BADL). According to the KEYNOTE-564 criteria, he met the definition of “high-risk” renal cell carcinoma (pT3a with sarcomatoid features and necrosis), a subgroup in which adjuvant pembrolizumab demonstrated consistent and clinically meaningful reduction in recurrence risk across all predefined age categories [3]. The decision to initiate adjuvant therapy was made in a multidisciplinary tumor board after assessing that his physiological reserve and functional capacity outweighed the chronological age. In this context, the balance between potential benefit and toxicity was considered acceptable, and immunotherapy was deemed appropriate following current international guideline recommendations for high-risk disease. This case, however, highlights the importance of individualized decision-making in older adults receiving immune checkpoint inhibitors, even when functional status is apparently optimal.

In our patient, HLH developed in temporal association with radiologic progression of clear cell renal carcinoma with sarcomatoid differentiation, a finding that may have contributed to the hyperinflammatory milieu and further complicated diagnostic interpretation. Persistent fever, pancytopenia, splenomegaly, marked hyperferritinemia (>49,000 ng/mL), hypofibrinogenemia, and elevated soluble IL-2 receptor (>7500 U/mL) fulfilled at least five of the eight HLH-2004 diagnostic criteria, confirming the diagnosis despite the absence of bone marrow biopsy (Table 3). Although these criteria were originally designed for pediatric populations, they remain a valuable framework when applied with clinical judgment. In adults, ferritin levels exceeding 10,000 ng/mL are considered highly suggestive of HLH, and elevated sCD25 confers strong diagnostic specificity (AUC ≈ 0.90). Complementary scoring systems such as the HScore can further support diagnosis; in this case, the calculated score indicated a high probability of HLH.

From a therapeutic standpoint, there are no standardized guidelines for ICI-associated HLH, and management is generally extrapolated from protocols for secondary forms (HLH-94 and HLH-2004), which include dexamethasone and etoposide. In older or frail adults, however, etoposide-related toxicity may limit its use, prompting consideration of targeted immunomodulators such as IL-1 blockade (anakinra), IL-6 inhibition (tocilizumab), or JAK inhibition (ruxolitinib) [18]. Our patient received high-dose corticosteroids followed by intravenous anakinra (100 mg every 6 h), but his condition rapidly deteriorated, leading to multiorgan failure and refractory shock. This fatal outcome aligns with previous reports, where overall mortality for ICI-associated HLH approaches 15%, despite prompt immunosuppressive therapy. This case underscores the importance of maintaining a high index of suspicion in patients treated with ICIs who develop unexplained fever, cytopenias, and hyperinflammatory markers, as early recognition and intervention remain crucial to improving outcomes. It also highlights the urgent need for specific diagnostic and therapeutic algorithms tailored to immune-mediated HLH in adults.

Our case represents, to our knowledge, the first reported fatal episode of pembrolizumab-induced HLH in adjuvant renal cell carcinoma, underscoring the need for heightened vigilance in this setting

## 4. Conclusions

Immune checkpoint inhibitor–associated hemophagocytic lymphohistiocytosis is a rare but highly morbid immune-related adverse event. Its nonspecific presentation—marked by persistent fever, cytopenias, and elevated inflammatory markers—can mimic infection, disease progression, or hepatic toxicity, making diagnosis particularly challenging. In patients receiving immunotherapy, HLH should always be included in the differential diagnosis when compatible clinical features are present. Early recognition and timely initiation of immunosuppressive therapy remain crucial to improving outcomes in this life-threatening condition.

## Figures and Tables

**Figure 1 reports-08-00256-f001:**
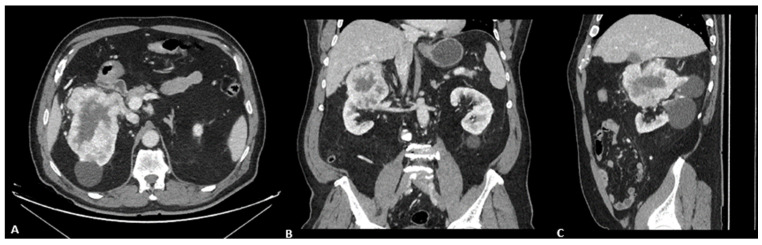
Contrast-enhanced abdominal CT in the portal phase with axial (**A**), coronal (**B**), and sagittal (**C**) reconstructions. In the upper third of the right kidney, there is a well-defined, multilobulated mass measuring approximately 8.3 × 9.6 × 6.6 cm (transverse, anteroposterior, and craniocaudal axes, respectively). The lesion shows intense peripheral enhancement with a central hypodense area consistent with necrosis. There are no signs of invasion into adjacent structures, although it lies in close contact with the inferior hepatic border, with no clear loss of the fat plane of separation.

**Figure 2 reports-08-00256-f002:**
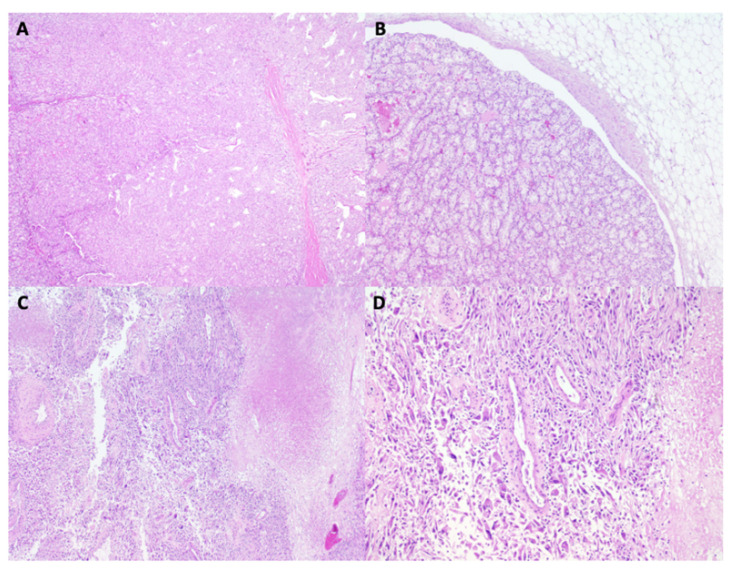
Histopathological features of the nephrectomy specimen (hematoxylin and eosin staining). (**A**) Low-power view showing the typical alveolar growth pattern of clear cell renal cell carcinoma, composed of polygonal cells with clear cytoplasm and distinct cell borders (×2). (**B**) Vascular invasion by clusters of neoplastic cells within a venous lumen (×20). (**C**) Area of coagulative tumor necrosis, characterized by extensive eosinophilic, acellular regions surrounded by viable tumor (×10). (**D**) Focus of rhabdoid differentiation, with large tumor cells displaying abundant eosinophilic cytoplasm, eccentric hyperchromatic nuclei, and prominent nucleoli (×20).

**Figure 3 reports-08-00256-f003:**
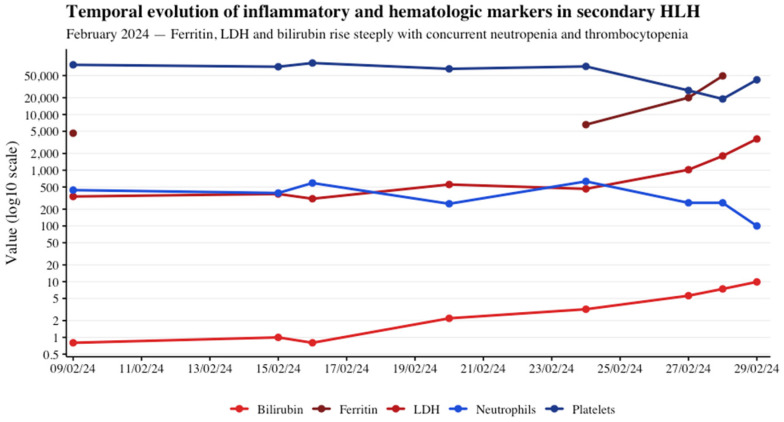
Temporal evolution of key inflammatory and hematologic markers in secondary HLH associated with pembrolizumab.

**Figure 4 reports-08-00256-f004:**
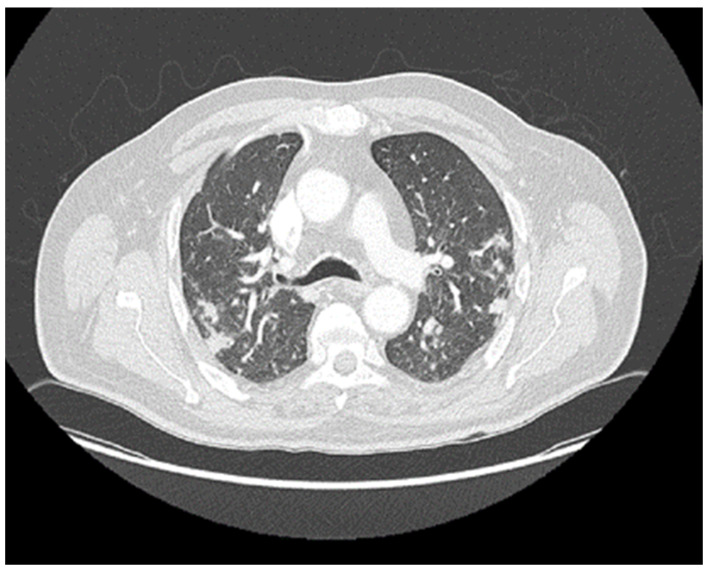
Chest CT scan in lung window performed in January 2024. The appearance of bilateral pulmonary nodules is observed, a finding consistent with progression of metastatic involvement.

**Table 1 reports-08-00256-t001:** Laboratory evolution of the patient from diagnosis (9 October 2023) to 48 h prior to death (29 February 2024).

DATE	10/08/23	17/01/24	06/02/24	09/02/24	15/02/24	16/02/24	20/02/24	24/02/24	27/02/24	28/02/24	29/02/24
Creatinine mg/dL	2.25	1.82	2.01	2.16	1.48	1.42	1.54	1.65	1.58	1.71	2.88
Glomerular filtration rate (GFR) ml/min/1.73 m^2^	27	34	30	28	44	46	42	39	41	37	20
LDH U/L	127	155	304	338	375	308	555	463	1021	1803	3640
Total bilirrubin mg/dL	0.8	0.5	0.8	0.8	1	0.8	2.2	3.2	5.6	7.4	9.9
Direct bilirrubin mg/dL	NA	NA	NA	NA	NA	NA	1.6	2.5	4.6	6.1	7.6
GGT U/L	194	435	428	337	680	545	729	716	521	542	482
Alkaline phosphatase U/L	137	227	153	127	209	166	230	223	197	240	281
Albumin g/dL	4.3	3.7	3.7	2.6	3.2	2.8	3.6	3.3	2.8	3	3.1
NTproBNP (pg/mL)	NA	NA	NA	NA	NA	NA	NA	NA	NA	NA	6984
C reactive protein (CRP)	NA	4.14	8.99	22.47	19.5	8.62	18.17	6.7	21.95	23.81	32.11
Procalcitonin (ng/mL)	NA	NA	0.31	NA	0.55	NA	NA	NA	NA	NA	7.29
Triglycerides (mg/dL)	134	NA	NA	NA	209	NA	NA	NA	NA	NA	NA
IL-6 (pg/mL)	NA	NA	NA	NA	NA	11.7	NA	NA	NA	NA	NA
Hemoglobin (g/dL)	13.1	10.7	12.9	10.7	10.1	8.8	9.4	9.7	8	8.9	9.6
Leukocytes /µL (×10^9^/L)	3290	2780	2140	580	720	810	460	840	380	480	440
Neutrophils /µL (×10^9^/L)	1740	2080	1500	440	390	590	250	630	260	260	100
Platelets /µL (×10^9^/L)	127,000	321,000	124,000	78,000	72,000	84,000	66,000	73,000	27,000	19,000	42,000
IL-2 (U/mL)	NA	NA	NA	NA	NA	NA	>7500	NA	NA	NA	NA
Ferritin (ng/mL)	239	NA	NA	4618	NA	NA	NA	6575	20,074	49,274	NA
Fibrinogen (mg/dL)	NA	NA	NA	NA	287.6	NA	NA	151.8	NA	233.8	NA

**Table 2 reports-08-00256-t002:** Published cases of pembrolizumab-associated hemophagocytic lymphohistiocytosis (HLH).

Author	Gender	Age	Pembrolizumab Cycles	Time to Onset	Line of Therapy	HLH Treatment	Tumor Type	Death
Kalmuk et al. (2020) [7]	Male	61	14	74 weeks	1	Methylprednisolone Dexamethasone 10 mg/m^2^ Etoposide 150 mg/m^2^	Squamous cell orofaringe	No. Immunotherapy was restarted.
J.Doyle et al. (2021) [10]	Male	71	14	42 weeks	1	NA	Lung adenocarcinoma	No. Switch to another line of therapy.
Zhai et al. (2024) [11]	Woman	73	1	1 week	1	Methylprednisolone, posaconazole, and caspofungin *	Squamous cell carcinoma cervix	No. Alive at 6 months.
Wei et al. (2022) [12]	Woman	50	1	1 week	2	Oral methylprednisolone and azathioprin	Thymic carcinoma	No. Alive at 14 months.
Wei et al. (2022) [12]	Male	70	1	2 weeks	1	Methylprednisolone Dexamethasone 10 mg/m^2^ Etoposide 150 mg/m^2^	Squamous cell lung	No. Alive at 8 months.
Takahasi et al. (2020) [13]	Male	78	1	1 week	2	Methylprednisolone (1000 mg/day for 3 days) After the pulse steroid therapy, he received 60 mg/day (1 mg/kg/day) of prednisolone, which was tapered to 50 mg/day within 4 weeks	Lung adenocarcinoma	No
Sadaat et al. (2018) [14]	Male	58	6	20 weeks	1	High-dose glucocorticoids, steroid dose was tapered over 7 weeks	Melanoma	No. Complete response for 1 year.
Al-Samkari et al. (2018) [15]	Woman	58	5	15 weeks	2	Methylprednisolone taper (initial dose 1 g, with slow taper over the following weeks) (No etoposide given due to observed improvement)	Breast cancer	No. Resolution of the condition. *PRF1A91V* gene polymorphism Included in Keynote NCT02513472
G.L olmes et al. (2025) [16]	Woman	32	11	33 weeks	1 (adyuvant)	Prednisolone orally 100 mg once daily	Breast cancer	No. Resolution of the condition.
Patton et al. (2024) [5]	Woman	40	4	12 weeks	1 (neoadyuvant)	Methylprednisolone 100 mg/day Tocilizumab (8 mg/kg), 2 doses	Breast cancer	No. Response to tocilizumab. Required intubation.
Akagi et al. (2020) [17]	Male	74	1	4 weeks	2	Dexamethasone 10 mg/m^2^ Etoposide 150 mg/m^2^	Lung adenocarcinoma	No.
Marar et al. (2022) [18]	Woman	80	6	22 weeks	1	1 mg/kg methylprednisolone dexamethasone 10 mg/kg tocilizumab at 4 mg/kg 2 doses	SCC	No. Resolution of the condition.
Honda et al. (2025) [19]	Male	63	1	2 weeks	1 (plus lenvatinib)	Methylprednisolone 1000 mg/day mycophenolate mofetil was initiated at 1000 mg/day and escalated to 2000 mg/ day over 3 days Ganciclovir (positive citomegalovirus due to immunosupresion)	Clear cell renal cell carcinoma	No Primary tumor resected after 6 months; died from pleural effusion 1 month after surgery.
Rossignon et al. (2024) [20]	Male	56	NE	It developed after discontinuation of pembrolizumab. The patient was receiving third-line therapy with enfortumab vedotin (E–V).	2	Dexamethasone Etoposide	Metastatic urothelial carcinoma	Yes. Patient death after 28 days of hospitalization.

* Caspofungin was administered because the Aspergillus test result was positive.

**Table 3 reports-08-00256-t003:** Diagnostic criteria for Hemophagocytic Syndrome. In the absence of bone marrow biopsy, at least 5 must be fulfilled.

Fever ≥ 38.5 °C
Splenomegaly
Two or more cytopenias (hemoglobin < 9 g/dL, neutrophils < 1000/µL, platelets < 100,000/µL)
Hypofibrinogenemia < 150 mg/dL and/or hypertriglyceridemia > 265 mg/dL
Hemophagocytosis in bone marrow, spleen, lymph node, or liver
Low or absent natural killer (NK) cell activity
Ferritin > 500 ng/mL
Elevated soluble CD25 (soluble interleukin-2 receptor alpha) > 2400 U/mL

## Data Availability

The data presented in this study are available on request from the corresponding author. The data are not publicly available due to patient privacy.

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
