# Peer review of "Pembrolizumab-Associated Hemophagocytic Lymphohistiocytosis in Clear Cell Renal Carcinoma: Case Report and Literature Review"

_reports, 2025, doi:10.3390/reports8040256_

Round 1
Reviewer 1 Report
Comments and Suggestions for Authors
- The treatment choice is dubious. Why did the researchers choose to give adjuvant therapy in early stage but high risk patient at age of 80 with serious comorbidities. Those must be extensively discussed in discussion section focusing on age , disease stage and comorbidities
Author Response
We thank the reviewer for this important comment. We agree that the rationale for choosing adjuvant pembrolizumab in an 80-year-old patient with comorbidities must be more explicitly discussed. We have added a detailed explanation addressing the patient’s excellent baseline functional status (ECOG 0, Karnofsky 100), his classification as a high-risk case according to the KEYNOTE-564 criteria, and the multidisciplinary decision-making process that balanced age, disease stage, and comorbidities. These clarifications have been incorporated between lines 219 and 233 of the revised manuscript.
Reviewer 2 Report
Comments and Suggestions for Authors
Dear authors,
This is a very good work, which describes a Pembrolizumab-associated hemophagocytic lymphohistiocytosis in clear cell renal carcinoma. The introduction is brief, but informative; the case is described in detail, with appropriate figures and tables. In the discussion, there is a table with all the cases of similar cases described so far, which helps the reader to look them up and compare them accordingly. The conclusion is understandable. The references are relevant and recent. I believe that this particular manuscript is worth publishing.
Author Response
We sincerely thank the reviewer for the positive and encouraging evaluation of our manuscript. We appreciate the recognition of the clarity of the introduction, the detailed case description, the quality of the figures and tables, and the comprehensiveness of our literature review. We are also grateful for the reviewer’s acknowledgement of the relevance and timeliness of the references included.
We are pleased that the reviewer considers the manuscript suitable for publication. Although no specific changes were requested, we have carefully revised the text to ensure clarity, consistency, and full alignment with the journal’s formatting standards.
We greatly appreciate the reviewer’s thoughtful comments and support.
Reviewer 3 Report
Comments and Suggestions for Authors
This paper was reported a case of hemophagocytic lymphohistiocytosis occurring as an ICI-related adverse event in a patient who received adjuvant pembrolizumab therapy following nephrectomy. This paper is highly interesting. The reviewer would like to suggest some critiques as follows.
- The authors should follow the journal style indicated in the instructions. Especially, the way the text and references are described is incorrect.
- The citation numbering format is incorrect.
- On line 22, HLH should be listed later in the sequence because it is an AE that occurred after pembrolizumab administration.
- On line 25 and 32, the diagnostic criteria for HLH should be presented.
- On line 55, the administration criteria for adjuvant pembrolizumab therapy should be documented.
- On line 60, the literature supporting this written statement should be cited.
- On line 73, what is R0? The authors should be described about clinical TNM.
- The presentation method for Tables 1 and 2 is inappropriate.
- Why is the text in bold in the Discussion section?
Author Response
This paper was reported a case of hemophagocytic lymphohistiocytosis occurring as an ICI-related adverse event in a patient who received adjuvant pembrolizumab therapy following nephrectomy. This paper is highly interesting. The reviewer would like to suggest some critiques as follows.
The authors should follow the journal style indicated in the instructions. Especially, the way the text and references are described is incorrect.
We appreciate this observation. We have revised the entire manuscript to fully comply with the journal’s formatting guidelines, including text style, section structure, and reference formatting.
The citation numbering format is incorrect.
We have corrected this issue. The problem arose from the use of a Spanish reference manager, but it has now been fully resolved. We have also reviewed the citation order and removed duplicated references.
On line 22, HLH should be listed later in the sequence because it is an AE that occurred after pembrolizumab administration.
We thank the reviewer for this observation. We agree that HLH should appear later in the sequence of events, as it occurred after the administration of pembrolizumab. The sentence has been revised accordingly to reflect the correct chronological order.
On line 25 and 32, the diagnostic criteria for HLH should be presented.
We thank the reviewer for this helpful suggestion. We have adjusted the sentence in the Abstract to clarify which HLH-2004 diagnostic criteria were fulfilled by the patient. Since the complete set of diagnostic criteria is presented in detail later in the manuscript (Table 3), we believe that expanding this information further in the Abstract is not necessary.
On line 55, the administration criteria for adjuvant pembrolizumab therapy should be documented.
We thank the reviewer for this comment. We have clarified the administration criteria for adjuvant pembrolizumab in high-risk clear cell renal cell carcinoma. Specifically, we added a sentence indicating that the patient met the eligibility criteria defined in the KEYNOTE-564 trial, which established pembrolizumab as the standard adjuvant therapy for patients with high-risk features after nephrectomy. The revised text has been added immediately after the sentence on line 55.
On line 60, the literature supporting this written statement should be cited.
We thank the reviewer for this observation. We agree that the statement on line 60 required supporting literature. We have now added an appropriate citation to strengthen this point, specifically the recent systematic review by Naleid et al. (2025), which provides detailed evidence regarding pembrolizumab-associated toxicity:
Naleid N, Mahipal A, Chakrabarti S. Toxicity Associated with Pembrolizumab Monotherapy in Patients with Gastrointestinal Cancers: A Systematic Review of Clinical Trials. Biomedicines. 2025;13(1):229.
This reference has been incorporated into the revised manuscript at the corresponding position.
On line 73, what is R0? The authors should be described about clinical TNM.
We thank the reviewer for this helpful clarification request. In the revised manuscript, we have defined the term R0 as a complete surgical resection with negative margins. Additionally, we have expanded the description of the patient’s pathological staging by explicitly documenting the full clinical and pathological TNM classification (pT3aN0M0) in the corresponding sentence. These modifications ensure that the terminology is clear and consistent with oncologic reporting standards.
The presentation method for Tables 1 and 2 is inappropriate.
We thank the reviewer for this comment. In response, we have reorganized Table 1 by removing analytical parameters that were not clinically relevant for the interpretation of the case, thereby improving clarity and readability. Additionally, we have revised Table 2 to ensure appropriate formatting and have carefully verified all references included in the table to avoid duplication and maintain consistency with the reference list. These adjustments have been incorporated in the revised manuscript.
Why is the text in bold in the Discussion section?
We thank the reviewer for pointing this out. The bold text in the Discussion section was unintentional and resulted from an internal drafting artifact during the collaborative preparation and translation of the manuscript, when certain passages were temporarily highlighted for emphasis among the authors. This formatting has now been fully removed in the revised version.
Round 2
Reviewer 3 Report
Comments and Suggestions for Authors
none.